# Learning Dynamic Query Combinations for Transformer-based Object Detection and Segmentation

## Abstract

Transformer-based detection and segmentation methods use a list of learned detection queries to retrieve information from the transformer network and learn to predict the location and category of one specific object from each query. We empirically find that random convex combinations of the learned queries are still good queries for the corresponding models. We then propose to learn a convex combination with dynamic coefficients based on the high-level semantics of the image. The generated dynamic queries better capture the prior of object locations and categories in the different images. Equipped with our dynamic queries, a wide range of DETR-based models achieve consistent and superior performance across multiple tasks (object detection, instance segmentation, panoptic segmentation) and on different benchmarks (MS COCO, CityScapes, YoutubeVIS).

## 1 Introduction

Object detection is a fundamental yet challenging task in computer vision, which aims to localize and categorize objects of interest in the images simultaneously. Traditional detection models (Ren et al., 2015; Cai & Vasconcelos, 2019; Duan et al., 2019; Lin et al., 2017b;a) use complicated anchor designs and heavy post-processing steps such as Non-Maximum-Suppression (NMS) to remove duplicated detections. Recently, Transformer-based object detectors such as DETR (Carion et al., 2020) have been introduced to simplify the process. In detail, DETR combines convolutional neural networks (CNNs) with Transformer (Vaswani et al., 2017) by introducing an encoder-decoder framework to generate a series of predictions from a list of object queries. Following works improve the efficiency and convergence speed of DETR with modifications to the attention module (Zhu et al., 2021; Roh et al., 2021), and divide queries into positional and content queries (Liu et al., 2022; Meng et al., 2021). This paradigm is also adopted for instance/panoptic segmentation, where each query is associated with one specific object mask in the decoding stage of the segmentation model (Cheng et al., 2021a).

The existing DETR-based detection models always use a list of fixed queries, regardless of the input image. The queries will attend to different objects in the image through a multi-stage attention process. Here, the queries are served as global priors for the location and semantics of target objects in the image. In this paper, we would like to associate the detection queries with the content of the image, i.e., adjusting detection queries based on the high-level semantics of the image in order to capture the distribution of object locations and categories in this specific scene. For example, when the highlevel semantics show the image is a group photo, we know that there will be a group of people (category) inside the image and they are more likely to be close to the center of the image (location).

Since the detection queries are implicit features that do not directly relate to specific locations and object categories in the DETR framework, it is hard to design a mechanism to change the queries while keeping them within a meaningful "query" subspace to the model. Through an empirical study, we notice that convex combinations of learned queries are still good queries to different DETR-based models, achieving similar performance as the originally learned queries (See Section 3.2). Motivated by this, we propose a method to generate dynamic detection queries based on the high-level semantics of the image in DETR-based methods while constraining the generated queries

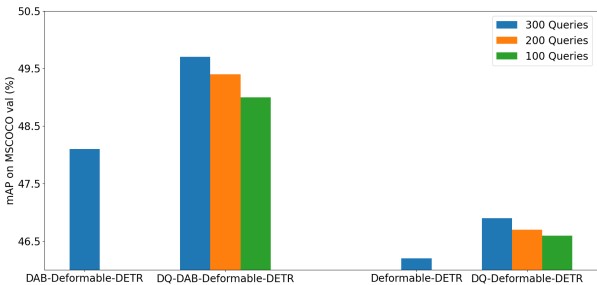

Figure 1: Comparison of DETR-based detection models integrated with and without our methods on MS COCO (Lin et al., 2014) `val` benchmark. ResNet-50 is used as the backbone.

in a sequence of convex hulls spanned by the static queries. Therefore, the generated detection queries are more related to the target objects in the image and stay in a meaningful subspace. Extensive experiments on MS COCO (Lin et al., 2014), CityScapes (Cordts et al., 2016) and YouTube-VIS (Yang et al., 2019) benchmarks with multiple tasks, including object detection, instance segmentation, and panoptic segmentation show superior performance of our approach combined with a wide range of DETR-based models. In Figure 1, we show the performance of our method on object detection combined with two baseline models. When integrated with our proposed method, the mAP of recent detection models DAB-Deformable-DETR (Liu et al., 2022) can be increased by 1.6%. With fewer dynamic detection queries and less computation in the transformer decoder, our method can still achieve better performance than baseline models on both Deformable-DETR and DAB-Deformable-DETR.

## 2 RELATED WORKS

**Transformers for object detection.** Traditional CNN-based object detectors require manually designed components such as anchors (Ren et al., 2015; Tian et al., 2019) or post-processing steps such as NMS(Neubeck & Van Gool, 2006; Hosang et al., 2017). Transformer-based detectors directly generate predictions for a list of target objects with a series of learnable queries. Among them, DETR (Carion et al., 2020) first combines the sequence-to-sequence framework with learnable queries and CNN features for object detection. Following DETR, multiple works were proposed to improve its convergence speed and accuracy. Deformable-DETR (Zhu et al., 2021) and Sparse-DETR (Roh et al., 2021) replace the self-attention modules with more efficient attention operations where only a small set of key-value pairs are used for calculation. Conditional-DETR (Tian et al., 2020) changes the queries in DETR to be conditional spatial queries, which speeds up the convergence process. Anchor-DETR (Wang et al., 2021b) generates the object queries using anchor points rather than a set of learnable embeddings. DAB-DETR (Liu et al., 2022) directly uses learnable box coordinates as queries which can be refined in the Transformer decoder layers. DINO (Zhang et al., 2022) and DN-DETR Li et al. (2022) introduce a strategy to train models with noisy ground truths to help the model learn the representation of the positive samples more efficiently. Recently, Group-DETR Chen et al. (2022) and HDETR Jia et al. (2022) both added auxiliary queries and a one-to-many matching loss to improve the convergence of the DETR-based models. They still use static queries which does not change the general architecture of DETR. All these Transformer-based detection methods use fixed initial detection queries learned on the whole dataset. In contrast, we propose to modulate the queries based on the image's content, which generates more effective queries for the current image.

**Transformers for object segmentation.** Besides object detection, Transformer-based models are also proposed for object segmentation tasks including image instance segmentation, panoptic segmentation (Kirillov et al., 2019; Wang et al., 2021a; Zhang et al., 2021) and video instance segmentation (VIS) (Yang et al., 2019). In DETR (Carion et al., 2020), a mask head is introduced on top of the decoder outputs to generate the predictions for panoptic segmentation. Following DETR, ISTR (Hu et al., 2021) generates low-dimensional mask embeddings, which are matched with the ground truth mask embeddings using Hungarian Algorithm for instance segmentation. SOLQ (Dong et al., 2021) uses a unified query representation for class, location, and object mask. Mask2Former

| Model | DAB-DETR | | Deformable-DETR | | Mask2Former |
|---|---|---|---|---|---|
| | $r = 2$ | $r = 4$ | $r = 2$ | $r = 4$ | $r = 2$ |
| Convex Combination | 37.9($\pm$0.10) | 30.4($\pm$0.20) | 35.0($\pm$0.20) | 24.2($\pm$0.05) | 41.2($\pm$0.10) |
| Non-convex Combination | 37.0($\pm$0.10) | 29.5($\pm$0.10) | 32.6($\pm$0.25) | 24.0($\pm$0.10) | 40.7($\pm$0.45) |
| Averaged Combination | 37.0 | 28.4 | 32.9 | 22.5 | 40.9 |
| Queries sampled randomly | 39.7($\pm$0.05) | 33.9($\pm$0.15) | 39.8($\pm$0.30) | 28.1($\pm$0.30) | 41.7($\pm$0.10) |

Table 1: Comparison of pretrained detection models DAB-DETR (Liu et al., 2022) and Deformable-DETR and segmentation model Mask2Former (Cheng et al., 2021a) with different queries. The shown metrics are box mAP for detection and mask mAP for segmentation. ResNet-50 is used as the backbone and models are evaluated on MS COCO `val`.

(Cheng et al., 2021a) introduces masked attention to extract localized features, and predict output for panoptic, instance and semantic segmentation in a unified framework. SeqFormer (Wu et al., 2021) utilizes video-level instance queries where each query attends to a specific object across frames in the video. These Transformer-based models follow the general paradigm of DETR and use fixed queries regardless of the input.

**Dynamic deep neural networks.** Dynamic deep neural networks (Han et al., 2021) aim at adjusting the computation procedure of a neural network adaptively in order to reduce the overall computation cost or enhance the model capacity. Slimmable networks (Yu et al., 2018; Yu & Huang, 2019; Li et al., 2021) introduce a strategy to adapt to multiple devices by simply changing channel numbers without the need for retraining. Dynamic Convolution (Chen et al., 2020) proposes a dynamic perceptron that uses dynamic attention weights to aggregate multiple convolution kernels based on the input features. On object detection, Dynamic R-CNN (Zhang et al., 2020) proposes a new training strategy to dynamically adjust the label assignment for two-stage object detectors based on the statics of proposals. Cui et al. (2022) proposes to train a single detection model which can adjust the number of proposals based on the complexity of the input image. Wang et al. (2021c) introduces a Dynamic Transformer to determine the number of tokens according to the input image for efficient image recognition, by stacking multiple Transformer layers with increasing numbers of tokens. In contrast to the existing work, we explore generating dynamic queries for a wide range of DETR-based models using the same framework. Our focus is not to reduce the computation cost of DETR-based models, but to improve the model performances with queries more related to the content of each individual image.

## 3 METHODOLOGY

### 3.1 PRELIMINARY

We first summarize the inference process of the existing Transformer-based models for a series of tasks, including object detection, instance segmentation, and panoptic segmentation, as the following Equation:

$$Y = \mathcal{N}_t \left( \mathcal{N}_{dec} \left( \mathcal{N}_{enc} \left( F \right), Q \right) \right).$$
(1)

For the object detection task, given the input image $I$, multi-scale features $F$ are extracted from the backbone network and then fed into a Transformer encoder $\mathcal{N}_{enc}$. After processing the features with multiple encoder layers, the output features are fed into a Transformer decoder $\mathcal{N}_{dec}$ together with $n$ randomly initialized query vectors $Q \in \mathbb{R}^{n \times f}$, where $n$ and $f$ denote the number of queries and length of each query respectively. Each query can be a feature vector (Carion et al., 2020; Zhu et al., 2021), or a learned anchor box (Liu et al., 2022). The outputs of $\mathcal{N}_{dec}$ is then fed into a task head $\mathcal{N}_t$ to generate the final predictions $Y = \{(b_i, c_i), i = 1, 2, \ldots, n\}$, where $b_i, c_i$ represent the bounding boxes and their corresponding categories of the detected objects. Then, the predictions are matched with the ground truths $Y^\star$ using the Hungarian Algorithm (Carion et al., 2020) to generate a bipartite matching. Then, the final loss is computed based on this bipartite matching:

$$\mathcal{L} = \mathcal{L}_{\text{Hungarian}} \left( Y, Y^\star \right).$$
(2)

For the segmentation tasks, the final predictions are updated to $Y = \{(b_i, c_i, m_i), i = 1, 2, \ldots, n\}$, where $m_i$ denotes the predicted masks for different object instances. Since there is no direct cor-

respondence of the predictions with the ground truth annotations, a bipartite matching is also computed to find the correspondence of the predictions and the ground truths $\boldsymbol{Y}^\star$. The final loss is then computed based on the matching. In some models such as Mask2Former (Cheng et al., 2021a), there will be no Transformer encoder $\mathcal{N}_{enc}$ to enhance the feature representations, while the other computational components follow the same paradigm.

## 3.2 FIXED QUERY COMBINATIONS

Though some existing works analyze the contents of the queries for the decoder, such as Conditional-DETR (Tian et al., 2020) and Anchor-DETR (Wang et al., 2021b), they always exam each query individually. To the best of our knowledge, there is no work studying the interaction between the queries in $\boldsymbol{Q}$. Here, we would like to explore what kind of transformations conducted between the learned queries still generate "good" queries. If we compute the average of a few queries, is it still an effective query? If we use different types of linear transformations, which would be better to produce good queries?

We conduct experiments to analyze the results of queries generated by different perturbations from the original queries. The procedure of the experiments is as follows: given a well-trained Transformer-based model, the initial queries for the decoder are denoted as $\boldsymbol{Q}^P = \left\{ \boldsymbol{q}_1^P, \boldsymbol{q}_2^P, \ldots, \boldsymbol{q}_n^P \right\} \in \mathbb{R}^{n \times f}$. The first type of perturbation uses linear combinations of the original queries. We first separate the $n$ queries into $m$ groups, where each group has $r = \frac{n}{m}$ queries and generates one new query. Then, we initialize the combination coefficients $\boldsymbol{W} \in \mathbb{R}^{m \times r}$, where $w_{ij} \in \boldsymbol{W}$ is the coefficient used for the $i$-th group, $j$-th queries, denoted as $\boldsymbol{q}_{ij}^P$, to generate a group of new queries $\boldsymbol{Q}^C = \{ \boldsymbol{q}_1^C, \boldsymbol{q}_2^C, \ldots, \boldsymbol{q}_m^C \} \in \mathbb{R}^{m \times f}$. The process can be summarized as:

$$\boldsymbol{q}_i^C = \sum_{j=1}^{r} w_{ij} \boldsymbol{q}_{ij}^P, \tag{3}$$

We use three settings to evaluate the impact of different coefficients in Equation 3, namely Convex Combination, Non-convex Combination, and Averaged Combination. In Convex Combination, $\boldsymbol{q}_i^C$ is within the convex hull of $\boldsymbol{q}_{ij}^P, j = 1, 2, \ldots, r$. The combination coefficients $w_{ij}$ are randomly initialized using uniform distribution in $[-1, 1]$ and then passed through a softmax function to satisfy the criteria: $w_{ij} \geq 0, \quad \sum_{j=1}^{r} w_{ij} = 1$. For Non-convex Combination, $w_{ij}$ are initialized in the same way as those in the convex combination, and the sum of $w_{ij}$ is forced to be 1. However, there is no guarantee on its range and $w_{ij}$ can be negative values. For Averaged Combination, we generate $\boldsymbol{q}_i^C$ by averaging $\boldsymbol{q}_{ij}^P, j = 1, 2, \ldots, r$. As a baseline, we evaluate the model on $m$ queries randomly sampled from $\boldsymbol{Q}^P$. The experiments are conduct on MS COCO benchmark (Lin et al., 2014) for object detection, and instance segmentation, using DAB-DETR (Liu et al., 2022), Deformable-DETR (Zhu et al., 2021) and Mask2Former (Cheng et al., 2021a), with ResNet-50 as the backbone. The results are summarized in Table 1. From Table 1, we notice that Convex Combination achieves the best results among all the compared settings except the baseline. Convex Combination only degenerate slightly compared with learned queries on DAB-DETR and Mask2Former. In addition, the performance of Convex Combination only has very small variances across different models, proving that convex combinations of the group-wise learned queries are naturally high-quality object queries for different Transformer-based models on both detection and segmentation tasks. $n$ is set to 300 for detection models and 100 for Mask2Former. We run each setting 6 times to compute the variance.

## 3.3 DYNAMIC QUERY COMBINATIONS

From the previous section, we learn that fixed convex combinations of learned queries are still able to produce a reasonable accuracy compared to the learned queries. In this section, we propose a strategy to learn dynamic query combinations for the Transformer-based models instead of randomly generating the coefficients $w_{ij}$ for query combinations. Our model predicts their values according to the high-level content of the input images. Therefore, each input image will have a distinct set of object queries fed into the Transformer decoder.

To generate dynamic queries, a naive idea is to generate the modulated queries directly from the input features $\boldsymbol{F}$. This method will increase the number of parameters dramatically, causing it difficult

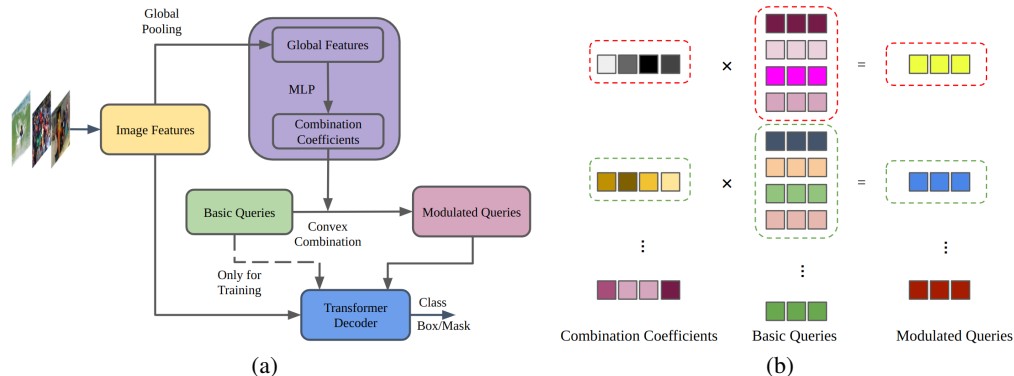

Figure 2: The framework of the proposed method. (a) Model pipeline with dynamic query combinations. The step with dashed line is only used in training. (b) Illustration of generating modulated queries from basic queries given combination coefficients.

to optimize and inevitably computationally inefficient. To verify this, we conduct an experiment on Deformable-DETR (Zhu et al., 2021) with ResNet-50 as the backbone. We replace the original randomly initialized queries with those generated by a multi-layer perceptron (MLP), which transforms the image feature $\boldsymbol{F}$ to $\boldsymbol{Q}$. With 50 epochs, the model only achieves $45.1\%$ mAP, which is lower than the original model with $46.2\%$.

Inspired by the dynamic convolution (Chen et al., 2020), which aggregates the features with multiple kernels in each convolutional layer, we propose a query modulation method. We first introduce two types of queries: basic queries $\boldsymbol{Q}^B \in \mathbb{R}^{n \times f}$ and modulated queries $\boldsymbol{Q}^M \in \mathbb{R}^{m \times f}$, where $n, m$ are the number of queries and $n = rm$. Equation 3 is updated as:

$$\boldsymbol{q}_i^M = \sum_{j=1}^{r} w_{ij}^D \boldsymbol{q}_{ij}^B, \tag{4}$$

where $\boldsymbol{W}^D \in \mathbb{R}^{m \times r}$ is the combination coefficient matrix and $w_{ij}^D \in \boldsymbol{W}^D$ is the coefficient for the $i$-th group, $j$-th query in $\boldsymbol{Q}^B$, denoted as $\boldsymbol{q}_{ij}^B$. To guarantee our query combinations to be convex, we add extra constraints to the coefficients as $w_{ij}^D \geq 0, \quad \sum_{j=1}^{r} w_{ij}^D = 1$.

In our dynamic query combination module, the coefficient matrix $\boldsymbol{W}^D$ is learned based on the input feature $\boldsymbol{F}$ through a mini-network, as:

$$\boldsymbol{W^D} = \sigma\left(\theta\left(\mathcal{A}\left(\boldsymbol{F}\right)\right)\right), \tag{5}$$

where $\mathcal{A}$ is a global average pooling to generate a global feature from the feature map $\boldsymbol{F}$, $\theta$ is an MLP, $\sigma$ is a softmax function to guarantee the elements of $\boldsymbol{W}^D$ satisfy the convex constraints. Here we try to make the mini-network as simple as possible to show the potential of using dynamic queries. This attention-style structure happens to be a simple and effective design choice.

During the training process, we feed both $\boldsymbol{Q}^M$ and $\boldsymbol{Q}^B$ to the same decoder to generate the corresponding predictions $\boldsymbol{Y}^M$ and $\boldsymbol{Y}^B$ as follows,

$$\boldsymbol{Y}^M = \mathcal{N}_t\left(\mathcal{N}_{dec}\left(\mathcal{N}_{enc}\left(\boldsymbol{F}\right), \boldsymbol{Q}^M\right)\right)$$
$$\boldsymbol{Y}^B = \mathcal{N}_t\left(\mathcal{N}_{dec}\left(\mathcal{N}_{enc}\left(\boldsymbol{F}\right), \boldsymbol{Q}^B\right)\right) \tag{6}$$

The final training loss is then updated to

$$\mathcal{L} = \mathcal{L}_{\text{Hungarian}}\left(\boldsymbol{Y}^M, \boldsymbol{Y}^\star\right) + \beta\mathcal{L}_{\text{Hungarian}}\left(\boldsymbol{Y}^B, \boldsymbol{Y}^\star\right) \tag{7}$$

where $\beta$ is a hyperparameter. During the inference, only $\boldsymbol{Q}^M$ is used to generate the final predictions $\boldsymbol{Y}^M$ while the basic queries $\boldsymbol{Q}^B$ are not used. Therefore, the computational complexity increases

| Backbone | Method | mAP | $AP_{0.5}$ | $AP_{0.75}$ |
|---|---|---|---|---|
| ResNet-50 | Conditional-DETR (Tian et al., 2020) | 40.9 | 61.7 | 43.3 |
| | DQ-Conditional-DETR | $42.0_{\uparrow 1.1}$ | $63.3_{\uparrow 1.6}$ | $44.2_{\uparrow 0.9}$ |
| | SMCA-DETR (Gao et al., 2021) | 41.0 | 61.5 | 43.5 |
| | DQ-SMCA-DETR | $42.1_{\uparrow 1.1}$ | $63.3_{\uparrow 1.8}$ | $44.9_{\uparrow 1.4}$ |
| | DAB-DETR (Liu et al., 2022) | 42.1 | 63.1 | 44.6 |
| | DQ-DAB-DETR | $43.7_{\uparrow 1.6}$ | $64.4_{\uparrow 1.3}$ | $46.6_{\uparrow 2.0}$ |
| | Deformable-DETR (Zhu et al., 2021) | 46.2 | 65.0 | 49.9 |
| | DQ-Deformable-DETR | $47.0_{\uparrow 0.8}$ | $65.5_{\uparrow 0.5}$ | $50.9_{\uparrow 1.0}$ |
| | DAB-Deformable-DETR (Liu et al., 2022) | 48.1 | 66.4 | 52.0 |
| | DQ-DAB-Deformable-DETR | $49.7_{\uparrow 1.6}$ | $68.1_{\uparrow 1.7}$ | $54.2_{\uparrow 2.2}$ |
| Swin-Base | Deformable-DETR (Zhu et al., 2021) | 50.9 | 70.5 | 55.3 |
| | DQ-Deformable-DETR | $53.2_{\uparrow 2.3}$ | $72.8_{\uparrow 2.3}$ | $57.7_{\uparrow 2.4}$ |
| | DAB-Deformable-DETR Liu et al. (2022) | 52.7 | 71.8 | 57.4 |
| | DQ-DAB-Deformable-DETR | $53.8_{\uparrow 1.1}$ | $72.8_{\uparrow 1.0}$ | $58.6_{\uparrow 1.2}$ |

Table 2: Comparison of existing DETR-based object detectors with/without our proposed methods integrated on MS COCO `val` split. ResNet-50 is used as the backbone.

for our models are negligible compared to the original DETR-based models. The only difference in the computation is that we have an additional MLP and a convex combination to generate the modulated queries. Since the role of modulated queries in our model is exactly the same as the fixed object queries in the original models, we use *modulated queries* and *queries* interchangeably to refer to the modulated queries.

## 4 EXPERIMENTS

To evaluate the effectiveness of our proposed methods, we first conduct experiments on a series of tasks, including object detection, instance segmentation, panoptic segmentation, and video instance segmentation with different DETR-based models. Then we conduct several ablation studies to investigate the impact of different hyperparameters in our model. Detailed experiment setups and visualization examples are provided in the supplementary materials.

### 4.1 EXPERIMENT SETUP

**Datasets.** For the object detection task, we use MS COCO benchmark (Lin et al., 2014) for evaluation, which contains $118,287$ images for training and $5,000$ for validation. For instance and panoptic segmentation, besides the MS COCO benchmark (80 "things" and 53 "stuff" categories), we also conduct experiments on the CityScapes (Cordts et al., 2016) benchmark (8 "things" and 11 "stuff" categories) to validate the effectiveness of our proposed method. For the VIS task, YouTube-VIS-2019 (Yang et al., 2019) is used for evaluation. For experiments on VIS, we pretrain our models on MS COCO and finetune them on the training set of YouTube-VIS-2019.

**Evaluation metrics.** For panoptic segmentation, the standard **PQ** (panoptic quality) metric (Kirillov et al., 2019) is used for evaluation. For instance segmentation (image or video) and object detection, we use the standard **mAP** (mean average precision) metric for evaluation. For VIS, mAP and AR (average recall) on video instances are the evaluation metrics.

**Implementation details.** The query ratio $r$ used to generate the combination coefficients is set to 4 by default. $\beta$ is set to be 1. $\theta$ is implemented as a two-layer MLP with ReLU as nonlinear activations. The output size of its first layer is 512, and that of the second layer is the length of $W^D$ in corresponding models. For detection models, we use 300 queries if not specified otherwise. For segmentation models, we use 50 queries for Mask2Former on image segmentation tasks, 100 queries for Mask2Former on VIS, and 300 queries for SeqFormer on VIS. ResNet50 is used as the backbone for different models.

| Methods | mAP | $AP_{0.5}$ | $AP_{0.75}$ |
|---|---|---|---|
| Mask R-CNN (He et al., 2017) | 35.4 | 56.4 | 37.9 |
| QueryInst (Fang et al., 2021) | 39.8 | 61.8 | 43.1 |
| Mask2Former (Cheng et al., 2021a) (50 queries) | 42.4 | 64.3 | 45.7 |
| DQ-Mask2Former (50 queries) | $43.2_{\uparrow0.8}$ | $65.2_{\uparrow0.9}$ | $46.7_{\uparrow1.0}$ |
| Mask2Former (100 queries) | 43.3 | 65.5 | 46.9 |
| DQ-Mask2Former (100 queries) | $44.1_{\uparrow0.8}$ | $66.5_{\uparrow1.0}$ | $47.4_{\uparrow0.5}$ |

Table 3: Comparison of existing instance segmentation approaches and DQ-Mask2Former on MS COCO `val` split. All models use ResNet-50 as the backbone.

| Methods | PQ | RQ | $PQ_{th}$ | $RQ_{th}$ |
|---|---|---|---|---|
| UPSnet (Xiong et al., 2019) | 42.5 | 52.5 | 48.6 | 59.6 |
| DETR (Carion et al., 2020) | 43.4 | 53.8 | 48.2 | 59.5 |
| Mask2Former (Cheng et al., 2021a) | 50.4 | 59.9 | 55.8 | 65.9 |
| DQ-Mask2Former | $51.1_{\uparrow0.7}$ | $60.6_{\uparrow0.7}$ | $56.9_{\uparrow1.1}$ | $67.1_{\uparrow1.2}$ |

Table 4: Comparison of existing panoptic segmentation approaches with DQ-Mask2Former on MS COCO `val` split with ResNet-50 as the backbone.

| Method | Panoptic | | | | Instance | |
|---|---|---|---|---|---|---|
| | PQ | RQ | $PQ_{th}$ | $RQ_{th}$ | mAP | $AP_{0.5}$ |
| Mask2Former | 60.3 | 73.2 | 50.6 | 63.0 | 36.7 | 60.9 |
| DQ-Mask2Former | $61.6_{\uparrow1.3}$ | $74.3_{\uparrow0.9}$ | $53.5_{\uparrow2.9}$ | $65.7_{\uparrow2.7}$ | $37.5_{\uparrow0.8}$ | $62.2_{\uparrow1.3}$ |

Table 5: Comparison of Mask2Former and DQ-Mask2Former on panoptic and instance segmentation tasks on CityScapes `val` split with ResNet-50 as the backbone.

## 4.2 MAIN RESULTS

**Object detection.** We evaluate our proposed methods with the DETR-based models Deformable-DETR (Zhu et al., 2021), SMCA-DETR (Gao et al., 2021), Conditional-DETR (Tian et al., 2020), DAB-DETR and DAB-Deformable-DETR (Liu et al., 2022) for object detection on the MS COCO benchmark. For a fair comparison, we run the original model integrated with and without our proposed dynamic queries using the same experimental settings, including the number of queries and epochs. The models equipped with our dynamic query combinations are denoted as DQ-Deformable-DETR, DQ-SMCA-DETR, DQ-Conditional-DETR, DQ-DAB-DETR, and DQ-DAB-Deformable-DETR, respectively. The results are shown as in Table 2. From Table 2, when integrated with our proposed method, mAP can be improved consistently by at least $0.8\%$ for all the models listed in the table. For DAB-Deformable-DETR, the mAP can be improved by $1.6\%$ with ResNet50 backbone and $1.1\%$ with Swin-B backbone. For Deformable-DETR, the mAP can be improved significantly by $2.3\%$ with Swin-B backbone. This proves the benefit of our method with different backbones. Note that models with dynamic queries only have negligible increased computation cost compared to the original models.

**Instance/panoptic segmentation.** Mask2Former(Cheng et al., 2021a) is a recent state-of-the-art model that can be used for different segmentation tasks with a unified model architecture. We compare Mask2Former with/without our dynamic queries for image instance and panoptic segmentation tasks on the MS COCO (Lin et al., 2014) and CityScapes (Cordts et al., 2016) benchmarks. The model plugged with dynamic queries is named DQ-Mask2Former. The results are shown as in Table 3, 4 and 5. We use 50 queries for all the listed settings. For instance segmentation (Table 3 and 5), our model DQ-Mask2Former achieves consistent improvement across different metrics compared to the original Mask2Former. For example, the performance on mAP is improved by around $0.8\%$ on both MS COCO and CityScapes. For panoptic segmentation, as shown in Table 4 and 5, DQ-Mask2Former again significantly outperforms Mask2Former (Cheng et al., 2021a) across all the evaluation metrics on both MS COCO and CityScapes.

| Method | mAP | $AP_{0.5}$ | $AP_{0.75}$ | AR |
|---|---|---|---|---|
| MaskTrack R-CNN (Yang et al., 2019) | 30.3 | 51.1 | 32.6 | 35.5 |
| IFC (Hwang et al., 2021) | 42.8 | 65.8 | 46.8 | 51.2 |
| Mask2Former (Cheng et al., 2021a) | 45.2 | 65.8 | 48.9 | 55.4 |
| DQ-Mask2Former | $46.3_{\uparrow 1.1}$ | $68.6_{\uparrow 2.8}$ | $50.6_{\uparrow 1.7}$ | $56.4_{\uparrow 1.0}$ |
| SeqFormer (Wu et al., 2021) | 46.0 | 68.5 | 50.4 | 53.6 |
| DQ-SeqFormer | $47.5_{\uparrow 1.5}$ | $70.3_{\uparrow 1.8}$ | $52.0_{\uparrow 1.6}$ | $55.0_{\uparrow 1.4}$ |

Table 6: Comparison of existing video instance segmentation approaches with DQ-Mask2Former and DQ-SeqFormer on YouTube-VIS-2019 `val` split with ResNet-50 as the backbone.

| $\beta$ | mAP | $AP_{0.5}$ | $AP_{0.75}$ | $AP_S$ | $AP_M$ | $AP_L$ |
|---|---|---|---|---|---|---|
| 0.0 | 45.6 | 64.1 | 49.4 | 27.2 | 49.1 | 60.5 |
| 0.5 | 46.4 | 65.0 | 50.3 | 28.1 | 49.2 | 62.6 |
| 1.0 | 47.0 | 65.5 | 50.9 | 28.8 | 50.1 | 62.2 |

Table 7: Analysis of $\beta$ using DQ-Deformable-DETR (ResNet-50 as the backbone) on the MS COCO benchmark with different settings.

**Video instance segmentation.** Besides image tasks, we also evaluate our method on the video instance segmentation task. We evaluated our method on two state-of-the-art VIS methods Mask2Former (Cheng et al., 2021b) and SeqFormer (Wu et al., 2021). Results are shown in Table 6. It can be seen from Table 6 that when integrated with our dynamic queries, mAP and AR of Mask2Former are improved by at around $1.0\%$. The mAP of SeqFormer is significantly boosted by $1.5\%$. Note the additional computation cost are negligible for these two models.

## 4.3 MODEL ANALYSIS

**Analysis of number of queries.** We use Deformable-DETR and DAB-Deformable-DETR as baseline models to study the effects of number of queries on the performance of object detection. We compare the baseline models with DQ-Deformable-DETR and DQ-DAB-Deformable-DETR integrated with different numbers of queries as in Figure 1. Note that we include the additional components of our models into the FLOPs computation of the decoder. When integrated with our method, even by reducing the number of queries from 300 to 100, the mAP of DQ-Deformable-DETR and DQ-DAB-Deformable-DETR are still better than the baseline models with 300 queries. We are also able to reduce the computation cost of the decoders of Deformable-DETR and DAB-Deformable-DETR by about $14\%$ and $24\%$ by using our method with 100 queries, respectively. However, we do not observe significant speedup using our method with fewer queries mainly because the main computation costs are from the backbones and the transformer encoders.

**Analysis of number of training epochs.** In Figure 3 (a), we show the impact of the number of training epochs on a sample model DQ-Deformable-DETR together with the original Deformable-DETR. From the figure, the mAP of DQ-Deformable-DETR is always better than that of the original Deformable-DETR at different epochs on the MS COCO benchmark. At early epochs around 30, DQ-Deformable-DETR achieves an even more significant performance gain compared then Deformable-DETR compared with later epochs.

**Analysis of $\beta$.** We analyze the impact of the scale of $\beta$ on models equipped with our dynamic queries. We conduct experiments using Deformable-DETR with ResNet-50 as the backbone on the MS COCO benchmark with different values of $\beta$. Results are shown in Table 7. As shown in the table, when $\beta$ is set to be 0, where no loss is directly computed with the prediction from the basic queries, the performance drops by $2.4\%$ compared to the original setting. In this case, the basic queries are not necessarily proper queries for the detection model, which will affect the quality of the modulated queries produced by them. The performance can be improved by increasing the value of $\beta$ to 0.5. Empirically we find $\beta = 1$ is a good choice to balance the scale of losses between the basic and modulated queries.

**Analysis of query ratio.** We use DQ-Deformable-DETR (Zhu et al., 2021) to analyze the performance of our proposed methods with different query ratios 2, 4, and 8, as in Figure 3 (b). From

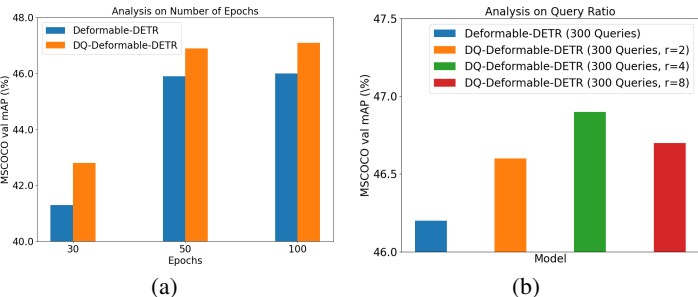

(a)          (b)

Figure 3: Analysis of the impact of number of epochs and query ratios on the performance.

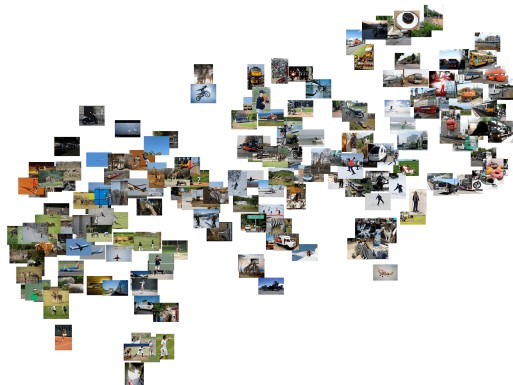

Figure 4: t-SNE visualization of $\boldsymbol{W}^D$ on 200 images from MS COCO `val`. Zoom in to see details.

the figure, using 4 as the query ratio achieves the best performance for DQ-Deformable-DETR with 300 queries. However, other query ratio choices still generate better accuracies than the original Deformable-DETR, which validates the effectiveness and robustness of our method.

**Visualization of $\boldsymbol{W}^D$.** Since $\boldsymbol{W}^D$ is conditioned on the high-level content of the image, we conjecture that images with similar scenes or object categories may have similar $\boldsymbol{W}^D$ parameters. We choose 200 images from the validation set of MS COCO and compute their $\boldsymbol{W}^D$ from DQ-DAB-DETR with 300 queries. The resulting $\boldsymbol{W}^D$ are first flattened into vectors, and then projected onto a two-dimensional space using t-SNE(Van der Maaten & Hinton, 2008). We visualize the projected $\boldsymbol{W}^D$ parameters along with their corresponding input images as Figure 4. We can see that some object categories tend to be clustered. For example, we can see a lot of transportation vehicles in the top right corner of the figure, and wild animals tend to be in the lower part of the figure, which indicates that the model uses some high-level semantics of the image to produce the combination coefficients.

## 5  CONCLUSION

In this paper, we propose to use dynamic queries depending on the input image to enhance DETR-based detection and segmentation models. We find that convex combinations of learned queries are naturally high-quality object queries for the corresponding models. Based on this observation, we design a pipeline to learn dynamic convex combinations of the basic queries, adapting object queries according to the high level semantics of the input images. This approach consistently improves the performance of a wide range of DETR-based models on object detection and segmentation tasks. The gain of our model is agnostic to the different designs of the Transformer decoders and different types of object queries. We believe this approach opens the door to designing dynamic queries and creates a new perspective for Transformer-based models.

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
