# OpenReview forum: "Learning Dynamic Query Combinations for Transformer-based Object Detection and Segmentation"
_ICLR.cc/2023/Conference — Submitted to ICLR 2023_

### Official Review · Reviewer_ykWm · 2022-10-17

**Confidence:** 5
**Correctness:** 3
**Technical Novelty And Significance:** 2
**Empirical Novelty And Significance:** 2
**Recommendation:** 5

**Clarity, Quality, Novelty And Reproducibility:**

> Clarity

Good.

> Quality

Good.

> Novelty

Relatively weak considering the recent HDETR and Group-DETR.

> Reproducibility

No code is available.

**Details Of Ethics Concerns:**

No further concerns.

**Strength And Weaknesses:**

> Strengths

✅ The presented idea is simple and easy to follow.

✅ The overall writing is satisfying.

✅ The proposed method can be used to improve performance across multiple detection and segmentation tasks.

> Weaknesses

❎ According to Table 2, the proposed method only slightly (+0.8) improves Deformable-DETR but improves DAB-Deformable-DETR significantly (+1.6), which is very STRANGE.
The authors explain why. Besides, the authors fail to compare their approach with a very important baseline, i.e., DN-DETR[1]. In fact, according to
the official implementation[2] of DN-DETR, DN-DAB-Deformable-DETR already achieves 49.5 while the proposed approach achieves 49.7 based on DAB-Deformable-DETR.
Although the proposed approach seems relatively different from DN-DETR/Group-DETR/HDETR, the authors are still encouraged to discuss the differences between the proposed approach with DN-DETR,
Group-DETR[3], and HDETR[4], which already verify that the key insight is to increase more positive samples during matching. If my understanding is correct,
the key is that **the proposed approach also benefits from adding more positive samples as each matched modulated query essentially corresponds to r=4 original queries, thus bringing 4x more positive samples**.

[1] DN-DETR: Accelerate detr training by introducing query denoising, CVPR2022

[2] https://github.com/IDEA-Research/DN-DETR

[3] Group-DETR: Fast DETR Training with Group-Wise One-to-Many Assignment, arXiv:2207.13085

[4] DETRs with Hybrid Matching, arXiv:2207.13080

❎ The experimental results are slightly WEAK and far from convincing as the authors only report improvements over the VERY weak results of Mask2Former and SeqFormer.
For example, the Mask2Former reports mAP=43.7/PQ=51.9 with R50 as the backbone while the authors only conduct experiments with a baseline of mAP=42.4/PQ=50.4 under a
short learning schedule. According to my experience, the proposed approach might become less useful under a longer training schedule. In summary, the experiments are WEAK and the authors should include more comparisons with results under the original learning schedule of Mask2Former or
SeqFormer. Besides, the authors should also report the comparison results with stronger backbones such as Swin-L to verify whether the proposed method generalizes well.

❎ Figure 1 is misleading to some degree considering the overall GFLOPs of the transformer decoder are much smaller compared to the transformer encoder
and backbone. In fact, the GFLOPs of the transformer encoder might be more than 10x larger if we take Deformable-DETR as an example. Therefore, the proposed
approach only brings VERY slight efficiency improvements.

❎ Some important details are missing:

- The authors should justify how to divide the basic queries into multiple groups. Especially, how to divide the basic queries seems non-trivial for a two-stage approach that selects the most confident queries to initialize the basic queries.

- **The authors are required to clarify how many queries are used during training indeed to avoid misunderstanding.**   As mentioned by the other reviewers, the authors seem to try to hide some very important details.

**Summary Of The Paper:**

This paper presents an interesting approach to improving DETR-based models across multiple tasks. The proposed approach mainly consists of the following three steps:
- learn image-adaptive dynamic coefficients for each of the original grouped learned queries,
- perform convex combination, i.e., aggregate, each group of learned queries according to coefficients, thus producing a small set of clustered queries, and
- apply Hungarian matching over the clustered queries and the original queries.

Last, the proposed approach also achieves relatively WEAK improvements across multiple benchmarks.
In general, the proposed approach seems to be capable of

- modeling the co-occurring relations between different objects, thus I guess the proposed approach
mainly benefits from explicitly modeling the interactions between different objects. However, the authors still use the original single-object distribution to
act as the training signal instead of constructing a set of multi-object distributions, which makes the proposed approach less reasonable.

- increasing more positive samples during matching. If my understanding is correct,
another key is that **the proposed approach also benefits from adding more positive samples as each matched modulated query essentially corresponds to r=4 original queries, thus bringing 4x more positive samples**, which is also verified in the recent approaches[1][2].

[1] DETRs with Hybrid Matching, arXiv:2207.13080

[2] Group-DETR: Fast DETR Training with Group-Wise One-to-Many Assignment, arXiv:2207.13085

**Summary Of The Review:**

The authors' efforts on pushing the frontier of object detection and segmentation performance along the DETR path are highly encouraging.
The overall idea is interesting and the results are encouraging but relatively WEAK.
Please carefully address the above-listed weaknesses. I will increase the ratings if the authors could well address these concerns, especially to verify the proposed method over strong backbones under longer training schedules following Mask2Former and SeqFormer.

---

> ### Author Response · Authors · 2022-11-17
> **Thanks for the review.**
>
> Thanks for the review. The followings are the replies.
>
> Q1: According to Table 2, the proposed method only slightly(+0.8) improves Deformable-DETR but improves DAB-Deformable-DETR significantly(+1.6), which is very STRANGE. The authors explain why.
>
> A1: It is totally normal for a generally applicable method to achieve more improvements on some tasks/backbones and lower improvements on a few other tasks/backbones. We do not think this is strange.
>
> Q2: Differences from DN-DETR, comparison with it.
>
> A2: DN-DETR has a different focus compared to ours. DN-DETR focuses on adding noisy ground truth box-label pairs as queries to help the model learn the representation of the positive samples more efficiently. Due to the specific design of DN-DETR, it can only be applied to specific DETR-based models which use bounding box coordinates or anchor points as queries (Although authors of DN-DETR claim their method is general, all of the experiments in their paper is based on DAB-DETR. Extending their method to the general DETR framework is non-trivial). Our method is much more general and can be adopted by most DETR-based models for detection, instance segmentation and panoptic segmentation. Due to the page limit, we do not think it is necessary to heavily compare our generally applicable method with a method that can only be applied to a specific type of model.
>
> Q3: Differences from Group-DETR, and HDETR
>
> A3: Group-DETR and HDETR added auxiliary queries and a one-to-many matching loss to improve the convergence of the DETR-based models. They still use static queries which do not change the model architecture of DETR. On the contrary, our method learns to dynamically adjust queries based on the high-level content of the input, which is a key improvement to the DETR framework in general. To study the effect of more queries in training, we also conducted experiments to show the effect of simply adding another group of queries to the DETR-based models, which only shows marginally better results than the baselines (See the following Table 1 for Reviewer LvX2). This proves that the major cause of the improvement is our dynamic queries, not the added positive samples. Meanwhile, both Group-DETR and HDETR are concurrent works, which just released on arxiv in July and August, close to the submission deadline of ICLR. Also, Group-DETR is a submission of ICLR this year. We have added a short discussion of these two methods in the updated draft.
>
> Q4: The experimental results are slightly weak. Results with stronger backbone.
>
> A4: It seems that the reviewer did not read the manuscript carefully. During the experiment section, we clearly explained that we use the same training schedule as the original models. We did not use a longer or shorter learning schedule. The reason that our results on Mask2Former are worse than the original paper is that we use 50 queries while the original paper uses 100. Nevertheless, we also added results for 100 queries in the updated draft (Also see Table 2 in the response for reviewer Qkjx). In terms of stronger backbones, we provide the results with Swin-B in Table 1 in the response for Reviewer Qkjx. Our method still achieves a significant gain with a more powerful backbone.
>
> Q5: Efficiency improvement.
>
> A5: We follow the advice and have modified the figure to remove the ambiguity. As discussed in the paper (See Section 4.3), our proposed model does not aim to improve efficiency. Compared with the existing counterparts, our model only has a negligible extra computation cost introduced.
>
> Q6: How to divide the basic queries into multiple groups.
>
> A6: We do not design any specific modules to group the basic queries on purpose. Since all the basic queries are randomly initialized, we group them in order the same as grouped convolutions.
>
> Q7: The authors are required to clarify how many queries are used during training indeed to avoid misunderstanding. As mentioned by the other reviewers, the authors seem to try to hide some very important details.
>
> A7: It seems that the reviewer did not notice the very important details already expressed in the original manuscript. The other reviewers do not have any questions about the number of queries for training. Please be more specific and read the other reviewers' comments more carefully if there are any additional concerns.  We have clearly introduced the settings on the number of queries in the paper (section 4.1): The query ratio $r$ used to generate the combination coefficients is set to $4$ by default. $\beta$ is set to be $1$. $\theta$ is implemented as a two-layer MLP with ReLU as nonlinear activations. The output size of its first layer is 512, and that of the second layer is the length of ${W}^D$ in corresponding models. For detection models, we use 300 queries if not specified otherwise. For segmentation models, we use 50 queries for Mask2Former on image segmentation tasks, 100 queries for Mask2Former on VIS, and 300 queries for SeqFormer on VIS.

---

### Official Review · Reviewer_QkjX · 2022-10-24

**Confidence:** 4
**Correctness:** 4
**Technical Novelty And Significance:** 2
**Empirical Novelty And Significance:** 3
**Recommendation:** 5

**Clarity, Quality, Novelty And Reproducibility:**

As discussed above, the illustration of this paper is clear and the overall quality for writing is acceptable. However, the originality of this paper might be a bit limited and its reproducibility cannot be guaranteed since the authors do not promise to release the code publicly in the future.

**Strength And Weaknesses:**

### Strength

**1. The idea is simple and sweet, which should be easy to follow.** The proposed method is essentially an attentional plug-in, which should be able to be implemented in just a few lines of code. Such simplicity will attract people to follow.

**2. The experiments are extensive and comprehensive.** The authors evaluate the proposed component into about 10 different frameworks for set-based object detection, instance and panoptic segmentation.

### Weaknesses

**1. The novelty is limited.** The proposed method is too similar to other attentional modules proposed in previous works [1, 2, 3]. The group attention design seems to be related to ResNeSt [4] but it is not discussed in the paper. Although these works did not evaluate their performance on object detection and instance segmentation, the overall structures between these modules and the one that this paper proposed are pretty similar.

**2. Though the improvement is consistent for different frameworks and tasks, the relative gains are not very strong.** For most of the baselines, the proposed methods can only achieve just about 1% gain on a relative small backbone ResNet-50. As the proposed method introduces global pooling into its structure, it might be easy to improve a relatively small backbone since it is with a smaller receptive field. I suspect whether the proposed method still works well on large backbone models like Swin-B or Swin-L.

**3. Some of the baseline results do not matched with their original paper.** I roughly checked the original Mask2former paper but the performance reported in this paper is much lower than the one reported in the original Mask2former paper. For example, for panoptic segmentation, Mask2former reported 51.9 but in this paper it's 50.4, and the AP for instance segmentation reported in the original paper is 43.7 but here what reported is 42.4.

Meanwhile, there are some missing references about panoptic segmentation that should be included in this paper [5, 6].

### Reference

[1] Chen, Yunpeng, et al. "A^ 2-nets: Double attention networks." NeurIPS 2018.

[2] Cao, Yue, et al. "Gcnet: Non-local networks meet squeeze-excitation networks and beyond." T-PAMI 2020

[3] Yinpeng Chen, et al. Dynamic convolution: Attention over convolution kernels. CVPR 2020.

[4] Zhang, Hang, et al. "Resnest: Split-attention networks." CVPR workshop 2022.

[5] Zhang, Wenwei, et al. "K-net: Towards unified image segmentation." Advances in Neural Information Processing Systems 34 (2021): 10326-10338.

[6] Wang, Huiyu, et al. "Max-deeplab: End-to-end panoptic segmentation with mask transformers." CVPR 2021

**Summary Of The Paper:**

This paper presents an attentioinal plug-and-play module, namely, Dynamic Query (DQ), to better generate object queries for set-based transformer detector and segmenter. The proposed method can consistently help improve different kinds of frameworks for both object detection, instance segmentation and panoptic segmentation.

**Summary Of The Review:**

Most of my concerns lie in the novelty issue and also the non-impressive performance. So I am inclined to reject this paper.

---

> ### Author Response · Authors · 2022-11-17
> **Thanks for the review.**
>
> Thanks for the comments. The followings are the details for the comments.
>
> Q1: The novelty is limited, similar to other attentional modules.
>
> A1: When designing our method, we try to make the mini-network to generate the modulated queries as simple as possible to show the potential of using dynamic queries. The attention-style structure happens to be a simple design choice. The novelty of our method is that we replace the static object queries with dynamic queries that are tuned by the content of the image in the DETR framework, not the structure of this mini-network.
>
> Q2: Though the improvement is consistent for different frameworks and tasks, the relative gains are not very strong.
>
> A2: We do not aim to set up the new state-of-the-art but to show how DETR-based methods can be further improved in general. Our method can be integrated with most of the existing DETR-based works and boost their performance for different tasks, including object detection, image instance/panoptic segmentation, and even video instance segmentation. It is totally normal for a generally applicable method to achieve significant improvements on some tasks and slightly lower improvements on a few other tasks.
>
> Q3: Results with other backbones.
>
> A3: We provide the experiment results with Swin-B as the backbone in Table 1, as suggested. Our method achieves consistent performance gain on Deformable-DETR, DAB-Deformable-DETR, and Mask2Former with Swin-B as the backbone.
>
> | Method                 | mAP                  | AP50                 | AP75                 |
> |------------------------|----------------------|----------------------|----------------------|
> | Deformable-DETR        | 50.9                 | 70.5                 | 55.3                 |
> | DQ-Deformable-DETR     | 53.2$_{\uparrow2.3}$ | 72.8$_{\uparrow2.3}$ | 57.7$_{\uparrow2.4}$ |
> | DAB-Deformable-DETR    | 52.7                 | 71.8                 | 57.4                 |
> | DQ-DAB-Deformable-DETR | 53.8$_{\uparrow1.1}$ | 72.8$_{\uparrow1.0}$ | 58.6$_{\uparrow1.2}$ |
> | Mask2Former | 46.7 | 70.2 | 50.4 |
> | DQ-Mask2Former | 47.3$_{\uparrow0.6}$ | 71.3$_{\uparrow1.1}$ | 51.1$_{\uparrow0.7}$|
>
> Table 1. Comparison of baselines and our proposed method on MS COCO val split. Swin-B is used as the backbone.
>
> Q4: Some of the baseline results do not match with their original paper.
>
> A4: Different from the settings in the original Mask2Former paper, which use $100$ queries by default, we use $50$ queries as mentioned in the manuscript. We also provide the experiment results with $100$ queries and compare them with the original paper, as Table 2. Although we use the same settings as the paper describes, the performance reproduced by us is slightly lower than the original paper. Compared to our reproduced version, DQ-Mask2Former achieved 0.8\% improvement on mAP, and 1.2\% improvement on AP50.
>
> | Method                   | mAP  | AP50 | AP75 | APS  | APM  | APL  |
> |--------------------------|------|------|------|------|------|------|
> | Mask2Former (Reproduced) | 43.3 | 65.3 | 46.7 | 23.0 | 46.2 | 64.5 |
> | Mask2Former (Original)   | 43.7 | 65.5 | 46.9 | 23.4 | 47.2 | 64.8 |
> | DQ-Mask2Former           | 44.1 | 66.5 | 47.4 | 23.7 | 47.5 | 66.0 |
>
> Table 2. Comparison on Mask2Former with $100$ queries on the MSCOCO benchmark for instance segmentation task.
>
> Q5: Missing references about panoptic segmentation.
>
> A5: Thank you for the suggestion. We have added them to the related works.

---

### Official Review · Reviewer_LvX2 · 2022-10-24

**Confidence:** 5
**Correctness:** 3
**Technical Novelty And Significance:** 3
**Empirical Novelty And Significance:** 3
**Recommendation:** 8

**Clarity, Quality, Novelty And Reproducibility:**

The paper is clearly written and easy to follow. The idea is novel as far as I know. I believe enough details are provided to allow reproduction.


Questions:
## Inference speed

Let's take an example to make sure I understand correctly. Let's say we start from a DeformableDETR model with 300 base queries, and we use r=4 as is done in the experiments. This will create 300/4 = 75 modulated queries. At train time, there are two forwards through the decoder, which naturally incurs a slightly higher cost. However, at inference time, only the modulated queries are used. So effectively the run-time is equivalent to a DeformableDETR with 75 queries plus the cost of computing the modulation coefficient. I would expect it to be faster because of the reduced computation in the decoder (quadratic self attention). However the paper seems to indicate that it's not really the case? Is it because computing the modulation coefficient is costly? Reporting some inference speeds would help here (eg Deformable DETR 300q vs Deformable DETR 75q vs DQ Deformable DETR 300q)

Typos:
- "visualization examples are proviede" (section 4)



**Strength And Weaknesses:**

Overall, the paper is clearly motivated, and shows improvements for a variety of models and tasks, suggesting the general applicability of the method.

## Non-modulated ablation

To me, the main weakness of the paper is that the ablations don't prove the usefulness of the *modulation* in itself.
In my opinion, here is the ablation that would demonstrate it convincingly:
Instead of adding $n/r$ *modulated* queries to the model (as is currently done), one could add the same number of *basic* queries. The rest of the design would be the same: this additional set of queries would be fed separately to the decoder and a separate matching would be computed for it.

The reason why this ablation is important is because it is known (see [1]) that adding extra groups of queries at training time with a separate matching helps with performance. Thus, the ablation above is required to demonstrate that the gains indeed come from the modulation component as claimed, and not from the same effect uncovered by [1].



[1] "Group DETR: Fast DETR Training with Group-Wise One-to-Many Assignment", Chen et al

**Summary Of The Paper:**

The paper proposes a general design improvement over models from the DETR family. Specifically, it introduces an additional set of *modulated queries* which are convex combinations of the regular queries. The weights used for the combination are predicted by a small MLP operating on the pooled representation of the image.
Then:
- At train time, both regular queries and modulated queries are trained, being fed independently to the decoder, and with a separate hungarian matching
- At test time, only the modulated queries are used.

The paper demonstrates the applicability of the method on a wide range of DETR variants as well as tasks (detection, instance segmentation, panoptic segmentation)

**Summary Of The Review:**

Without the result of the ablation I mentioned, it is not possible for me to conclude whether the reported improvements come from previously discovered phenomenon (ie using several groups of queries help performance) or from the claimed contribution of this paper.
As such, I recommend rejection for now, but depending on the result of the ablation I'm willing to increase my score.


POST REBUTTAL:
Authors have provided an ablation that better demonstrate their points. I'm increasing my score 5->8.

---

> ### Author Response · Authors · 2022-11-17
> **Thanks for your review**
>
> Thanks for your review and your kind help to make our paper better! We provide detailed replies to your questions.
>
> Q1: Non-modulated ablation.
>
> A1: Thanks for the great suggestion. We follow the reviewer's suggestion and train the model with two unrelated groups of queries with Deformable-DETR (300 queries) and Mask2Former (100 queries), as Table 1. The first group has the number of queries as our base queries while the second group has the same number of queries as our modulated queries. The second group is used in inference. From the table, with the two unrelated groups, there is still an improvement on the performance. However, our proposed method with dynamic queries achieves better results on the different models, which validates the effectiveness of our method.
>
> | Method                                            | mAP  | AP50 | AP75 |
> |---------------------------------------------------|------|------|------|
> | Deformable-DETR                                   | 46.2 | 65.0 | 49.9 |
> | DQ-Deformable-DETR                                | 47.0 | 65.5 | 50.9 |
> | Deformable-DETR (two groups of unrelated queries) | 46.5 | 64.9 | 50.5 |
> | Mask2Former                                       | 43.3 | 65.3 | 46.7 |
> | DQ-Mask2Former                                    | 44.0 | 66.5 | 47.4 |
> | Mask2Former (two groups of unrelated queries)     | 43.2 | 65.2 | 46.7 |
> Table 1. Comparison with two groups of unrelated queries
>
> Q2: Inference speed.
>
> A2: It is not costly to compute the modulation coefficients since it only uses a tiny network. The inference speed does not change much because the transformer decoder only has a small computation cost compared to the backbone and the Transformer encoder. We follow the reviewer's suggestion and post the inference speeds in Table 2. From the table, we notice there is not too much difference in the inference speed when changing the number of queries and our DQ-Deformable-DETR has a similar inference speed as the original model with the same settings.
>
> | Model                                                          | FPS  |
> |----------------------------------------------------------------|------|
> | Deformable-DETR (300 queries)                                  | 13.3 |
> | Deformable-DETR (75 queries)                                   | 14.0 |
> | DQ-Deformable-DETR (300 basic queries, 75 modulated queries)   | 13.5 |
> | DQ-Deformable-DETR (1200 basic queries, 300 modulated queries) | 13.1 |
>
> Table 2. Comparison of inference speeds of Deformable-DETR and DQ-Deformable-DETR with multiple settings. The backbone is ResNet-50 and the inference speed is tested with one TITAN RTX GPU.
>
> Q3: Typos.
>
> A3: Thanks for pointing that out. We have fixed the typos in the updated draft.

---

### Official Review · Reviewer_tEAE · 2022-10-24

**Confidence:** 4
**Clarity, Quality, Novelty And Reproducibility:** Good writing and interesting idea.
**Correctness:** 3
**Technical Novelty And Significance:** 3
**Empirical Novelty And Significance:** 3
**Recommendation:** 5

**Strength And Weaknesses:**

strength:
1. extensive experiments on various of tasks, showing effect of the proposed method.
2. the idea of convex combination of object queries is interesting, and the paper is with good motivation.
3. paper writing is good with clear structure.


weakness:

1. When dividing n queries into m groups, is the dividing process fully random? Is there better grouping strategy to discuss?
2. What kind of objects will the modulated queries correspond to during training and testing? What kind of objects are being improved? Is there visualization on the dynamic attention weights to help explain the improvement? Is there explanation on why non-convex combination is better than convex one?
3. The improvement in Table 3 and 4 is very limited.
4. why mapping one query to one group is necessary? can one query correspond to multiple modulated queries?

**Summary Of The Paper:**

The paper proposes a learnable convex combination of the learned object queries. High-level semantics are used to generate dynamic coefficients for query combination. Experiments are performed on a wide range of detr-based methods. Consistent improvements are observed on coco, cityscapes and ytvis.

**Summary Of The Review:**

Exploring the queries properties is meaningful; the experiment improvement shown on various datasets; the analysis can be further improved (see weakness).

---

> ### Author Response · Authors · 2022-11-17
> **Thank you for the review.**
>
> Thanks for your review and your kind help. We provide detailed replies to your questions.
>
> Q1: When dividing $n$ queries into $m$ groups, is the dividing process fully random? Is there a better grouping strategy to discuss?
>
> A1: We did not introduce any specific grouping strategy on purpose since the basic queries are equivalent to random initialization.
> We simply divide them in order. It would be meaningless to divide them using another grouping process but still with random initialization. There might be better grouping strategies by learning the queries and dividing progressively during training, but it is out of the scope of this work.
>
> Q2: What kind of objects will the modulated queries correspond to during training and testing? What kind of objects is being improved? Is there visualization on the dynamic attention weights to help explain the improvement?
>
> A2: The modulated queries will be mapped to all object instances during training, the same as those in the existing DETR-based methods. During inference, the modulated queries seem to pay more attention to the objects with a larger size, as shown in Table 1. Comparing AP$_\text{S}$, AP$_\text{M}$, and AP$_\text{L}$ for small, medium, and large objects, our models gain more improvement on larger objects on most model variants. In Figure 4 of the paper, we provide a visualization based on t-SNE to show the dynamic attention weights with different input images. Please refer to the paper for explanations.
>
> | Method                    | mAP   | AP50 | AP75 | APS   | APM   | APL   |
> |---------------------------|-------|--------|---------|-------|-------|-------|
> | Conditional\-DETR         | 40\.9 | 61\.7  | 43\.3   | 20\.6 | 44\.3 | 59\.3 |
> | DQ\-Conditional\-DETR     | 42\.0 | 63\.3  | 44\.2   | 21\.4 | 45\.6 | 60\.9 |
> | SMCA\-DETR                | 41\.0 | 61\.5  | 43\.5   | 21\.9 | 44\.3 | 59\.1 |
> | DQ\-SMCA\-DETR            | 42\.1 | 63\.3  | 44\.9   | 22\.6 | 45\.6 | 60\.3 |
> | DAB\-DETR                 | 42\.1 | 63\.1  | 44\.6   | 21\.5 | 45\.6 | 60\.3 |
> | DQ\-DAB\-DETR             | 43\.7 | 64\.4  | 46\.6   | 24\.4 | 47\.4 | 62\.4 |
> | Deformable\-DETR          | 46\.2 | 65\.0  | 49\.9   | 28\.3 | 49\.2 | 61\.5 |
> | DQ\-Deformable\-DETR      | 47\.0 | 65\.5  | 50\.9   | 28\.7 | 50\.1 | 62\.2 |
> | DAB\-Deformable\-DETR     | 48\.1 | 66\.4  | 52\.0   | 31\.4 | 51\.3 | 63\.3 |
> | DQ\-DAB\-Deformable\-DETR | 49\.7 | 68\.1  | 54\.2   | 31\.4 | 52\.7 | 65\.4 |
>
> Table 1. Comparison of existing DETR-based object detectors with/without our proposed methods integrated on MS COCO val split. ResNet-50 is used as the backbone.
>
> Q3: Is there an explanation for why a non-convex combination is better than a convex one?
>
> A3: First, a convex combination is better than a non-convex combination, not the other way around. We mainly learn the advantage of the convex combination from experiments. We conjecture that convex combination queries are closer to the original queries than those from non-convex combinations, and they form a high-quality “query subspace” to the model.
>
> Q4: The improvement in Table 3 and 4 is very limited.
>
> A4: Our method can be integrated with most of the existing DETR-based works and boost their performance for different tasks including object detection, image instance/panoptic segmentation and even video instance segmentation. It is totally normal for a generally applicable method to achieve significant improvements on some tasks and slightly lower improvements on a few other tasks. With the consistent performance gain across different models, datasets, and tasks, our method has shown its value and could be further extended for other tasks.
>
> Q5: Why mapping one query to one group is necessary? Can one query correspond to multiple modulated queries?
>
> A5: One query can be mapped to multiple groups.  Our design is mainly for simplicity. We also conduct experiments and provide more results where one basic query corresponds to two modulated queries. Specifically, we let one group of basic queries generate two modulated queries in Eq. (4) in the paper. We experiment with Deformable-DETR using ResNet-50 as the backbone. The number of modulated queries is $300$ and $r$ is set to $4$.  Results are shown in Table 2. From the table, we notice that when one basic query is mapped to two modulated queries, the performance slightly drops. A potential reason is that generating multiple queries in one group may harm the diversity of the queries, leading to degenerated performance.
>
> | Model                                                | mAP  | AP50 | AP75 |
> |------------------------------------------------------|------|------|------|
> | Deformable-DETR                                      | 46.2 | 65.0 | 49.9 |
> | DQ-Deformable-DETR (One group generates one query)   | 47.0 | 65.5 | 50.9 |
> | DQ-Deformable-DETR (One group generates two queries) | 46.8 | 65.3 | 50.7 |
>
> Table 2. Comparison of different ways to map basic queries to modulated queries.

---

### Author Response · Authors · 2022-11-19
**Summary of revision**

We are grateful for the comments and suggestions the reviewers provided to help us improve the quality of our manuscript. Besides replying to the comments point by point, we modified the submitted manuscripts with blue font color following the suggestions, summarized as:

1. We discussed and compared the recent works like DINO, DN-DETR, Group-DETR, and HDETR in the related works to show the differences between their models and our proposed method.

2. We discussed the panoptic segmentation works as reviewer QkjX recommended in the related work section.

3. We add experiment results using Swin-B as the backbone to show the consistent improvement of our proposed method.

4. We add the experiments with $100$ queries on Mask2Former to address the concerns of reviewer QkjX.

5. We fixed the typo, as reviewer LvX2 mentioned.

---

### Decision · Program_Chairs · 2023-01-20

**Decision:**

Reject

**Justification For Why Not Higher Score:**

The paper appears to have miss-characterized the results. The current comparisons are not completely fair and need to be redone in more comparable manner.

**Justification For Why Not Lower Score:**

N/A

**Metareview: Summary, Strengths And Weaknesses:**

The paper proposes an idea of learned convex combination queries for DETR architectures. The coefficients of these convex combination queries is conditioned on the individual images. The paper was reviewed by four reviewers and initially received more positive ratings, but after rebuttal the ratings have came down to:

* 1 x accept, good paper
* 3 x marginally below the acceptance threshold

Reviewers have raised a number of concerns with the work in rebuttal: (1) limited improvements [tEAE], (2) limited novelty [QkjX], and (3) lack of clear comparison to other methods with respect to the number of queries used. The last point is very serious, as nearly all reviewers initially believed that the number of queries in the proposed method were significantly lower than in reality. When this came to light, this raised serious concerns of multiple reviewers about the fairness of experiments and validity of results. If the number of queries is larger, then it is not at all clear that the proposed convex query combination is what is driving the improved performance.

AC has carefully looked at the reviews, rebuttal, discussion and the paper itself, and ultimately agrees with reviewers that this is problematic and, even if not done intentionally, is misleading. Notably, even after revisions this was not clarified in the revised manuscript. No revision mentions that authors "use 1200 basic queries and 300 modulated queries in most experiments", the text still reads that they use "300 queries". As such and in the current form the paper cannot be accepted for publication in ICLR.

Authors are encouraged to test more apples-to-apples (fairer) settings proposed by the reviewers, as well as to adjust tables (e.g., Table 2 and others) to include the number of queries and kinds used in each case, before resubmission to a future venue.